# Genetic Aspects of Corneal Sequestra in a Population of Persian, Himalayan and Exotic Cats

**DOI:** 10.3390/ani12152008

**Published:** 2022-08-08

**Authors:** Tom Schipper, Goedele Storms, Gerlinde Janssens, Sabine Schoofs, Eveline Capiau, Dieter Verdonck, Pascale Smets, Luc J. Peelman, Bart J. G. Broeckx

**Affiliations:** 1Department of Veterinary and Biosciences, Ghent University, B-9820 Merelbeke, Belgium; 2Veterinary Practice Kleidal, Kleidaallaan 74, B-2620 Hemiksem, Belgium; 3Veterinary Practice Clos Fleuri, Oude Gaversesteenweg 35, B-9820 Merelbeke, Belgium; 4Veterinary Practice Eveline Capiau, Ten Eekhovelei 239, B-2100 Deurne, Belgium; 5Small Animal Department, Ghent University, B-9820 Merelbeke, Belgium

**Keywords:** corneal sequestrum, heritability, felis catus, GWAS, breeding advice

## Abstract

**Simple Summary:**

Corneal sequestrum is a common eye disease in Persian, Himalayan and exotic cats. It is a brown stain on the cornea that can be painful and often requires surgery for full recovery. The cause of this disease is unknown. One hypothesis is that genetic factors influence the disease, but there are no data to support this in the scientific literature. In this study, the influence of genetics on this disease was investigated on the basis of clinical and pedigree data from a cattery. Kittens of some dams in this cattery turned out to have a higher risk of corneal sequestra than kittens of other dams, indicating that genetics indeed play a role. The disease also has a high heritability, which implies that by selecting the right breeding animals, breeders can reduce the risk of corneal sequestra in the offspring. DNA analysis of some of the cats from the cattery could not link the disease to a specific region in the DNA, so it is possible that many different genes in different DNA regions are involved in the development of corneal sequestra.

**Abstract:**

Corneal sequestra are ophthalmic lesions that are remarkably common in Persian, Himalayan and exotic cats. In this study, the genetic aspects of this disease were investigated in a population of cats originating from a single cattery. Odds ratios were calculated for parents with affected offspring. The heritability of (owner-reported) corneal sequestra was estimated with a Markov chain Monte Carlo procedure. Well-phenotyped cases and controls were used for a genome-wide association study. Data from 692 cats originating from the cattery, of which 61 were affected, were used. Cats from two specific mothers had significantly higher odds of developing corneal sequestra, but no significant effect of the fathers was found (after correction for the mothers). The heritability of corneal sequestra was estimated to be 0.96. A genome-wide association study with 14 cases and 10 controls did not reveal an associated chromosomal region. The large effect that genetic factors had on the development of corneal sequestra in this study suggests that selective breeding could be an effective way to reduce the prevalence of this condition in these cat breeds.

## 1. Introduction

Corneal sequestrum is an ophthalmic condition that is virtually unique to cats, as only a few cases have been described in horses [1] and dogs [2]. It is characterized by a local degeneration and brown—black discoloration of the cornea of variable size and severity. In mild cases, the sequestrum can spontaneously slough and heal, while more severe cases can persist for years or even progress to corneal perforation [3]. The level of discomfort varies from no signs of pain to extreme discomfort with epiphora, blepharospasm and photophobia [4]. In mild cases, the treatment can be limited to analgesia, but many sequestra are so deep and painful that they require surgical removal [3]. Surgical removal of a sequestrum consists of a keratectomy with or without the use of a grafting material depending on the depth of the sequestrum.

The age of onset varies from a few months to 17 years, but most cases occur between two and seven years [4]. Both sexes are affected equally. A breed predisposition in Persian cats and the related Himalayan and Exotic breed has been widely reported [3,5,6,7]. These breeds also have a higher risk of being bilaterally affected [3,5]. Other breeds reported to be predisposed are the Siamese, Birman and Burmese [8,9]. A prevalence of 2.4% (28 out of 1161) was found in a population of cats presented to a veterinary teaching hospital, making it the most common presumed hereditary or breed-related ocular disease in this population [10].

The pathogenesis of corneal sequestra remains mostly unclear. Trauma and infections, most notably Feline Herpesvirus I (FHV-1), have been proposed as etiologies affecting all breeds [3,4,11]. To explain the breed predisposition in brachycephalic cats, chronic irritation due to conformational factors such as entropion, trichiasis and lagophthalmos have been suggested [3,7,12]. Other possible explanations are breed-specific lower corneal sensitivity [13] and defects in the tear film [14], metabolism [4] or the corneal epithelium [15]. An autosomal recessive pattern of inheritance has been suggested [16], but no pedigree data have been published to support this. The aim of this study was to investigate the influence of the parents, heritability and association with genomic regions of corneal sequestra on the basis of a large pedigree.

## 2. Animals, Materials and Methods

### 2.1. Data Collection

All data and samples originated from a single cattery of Persian, Himalayan and exotic shorthair cats. The date of birth, sex, sequestrum status (affected or not) and parentage of all kittens, born between 2002 and 2021, were provided by the breeder, as well as the pedigrees of the breeding animals. The offspring was classified as affected if the breeder was informed by the owners that the cat was diagnosed with a sequestrum or had undergone surgical treatment for a corneal sequestrum. The offspring was classified as healthy if the breeder was informed by the owners that the cat was healthy or if no information on corneal sequestra was available.

### 2.2. Parent Risk Analysis

For each dam with affected offspring, the proportion of affected offspring was compared to that proportion among the total offspring of all other dams. Offspring born after 2019 were excluded from this analysis, as the youngest affected animals were born in 2019. An odds ratio and its confidence interval were calculated for each dam and Fisher’s exact test was used to determine statistical significance. The same procedure was followed for the three sires.

As the dams with the highest proportion of affected animals among their offspring had all been mated to the same sire, the effect of the sire could be confounded by the effect of the dams. To allow correction for this confounding, the dams were transformed to a binary variable based on the analysis above: high risk (odds ratio > 1 and *p* < 0.05) and low risk (odds ratio < 1 and/or *p* > 0.05). A logistic regression model was fit for the full dataset with sire, dam (as a binary variable) and year of birth as predictive variables for corneal sequestra. Significance of the covariates was determined with the likelihood ratio test.

### 2.3. Heritability Estimate

The heritability of corneal sequestra was estimated on the basis of the phenotypic data from the kittens born in the cattery and their ancestry data available from the pedigree of the dams and sires. The phenotype was modelled as the outcome of a logistic regression model containing the year of birth, sex and estimated breeding value of the animal and a random residual contribution. The estimated breeding value was computed as the additive genetic standard deviation, multiplied by the Mendelian sampling term and a random gametic effect, plus the average estimated breeding value of the parents. The Mendelian sampling term was calculated as the square root of the diagonal of the additive relationship matrix constructed with the optiSel package. The heritability was estimated as the additive genetic variance divided by the additive genetic variance plus the residual variance, which is assumed to be 1 for a binary trait [17].

The values of the parameters were estimated via a Bayesian approach using Markov chain Monte Carlo in the Stan programming language, as described by Cai et al. [18]. The priors for the intercept, year and sex coefficient were drawn from a normal distribution with mean zero and standard deviation 4 (normal (0, 4)) and the prior for the residual error from normal (0, 1). To avoid bias from the choice of the prior value of the additive genetic standard variation, four different distributions were used to draw priors from: a Cauchy distribution with mean 0 and standard deviation 2.5 (Cauchy (0, 2.5)), as suggested by Cai et al. [18], Cauchy (0, 10), normal (0, 10) and uniform (0, 20). Only positive values from these distributions were used. For each prior distribution, four parallel chains were run for 60,000 iterations. The first 30,000 of these were discarded as burn-in and every 30th value was sampled from the next 30,000 iterations. Examples of the Stan scripts and the used datasets are included as Appendix A.

### 2.4. Genome-Wide Association Study (GWAS)

A subset of the population was used for a genome-wide association study to identify chromosomal locations associated with corneal sequestra. Cases were defined as cats that had undergone surgery for a corneal sequestrum or were diagnosed with a corneal sequestrum on eye exam at the time of sampling. Controls were defined as cats that were at least two years old and declared free of corneal sequestra on eye exam at the time of sampling. Eye exams were performed by a board-certified veterinary ophthalmologist, with a certificate in veterinary ophthalmology, or a veterinarian with experienced ophthalmology knowledge. Affected animals suspected of FHV-1 infection based on the clinical history were excluded as cases for the GWAS.

DNA was extracted from whole blood on EDTA by the phenol-chloroform method [19]. The DNA was quantified with the Qubit 4.0 fluorometer (Invitrogen, Carlsbad, CA, USA) and at least 500 ng from each sample was used for genotyping on the Infinium iSelect 63k Cat DNA SNP genotyping array (Illumina, San Diego, CA, USA) [20].

Data from the GWAS were analyzed in PLINK v1.90b6.24 [21]. Variants and animals with a genotyping rate below 95% were excluded from the analysis, as were variants with more than one Mendelian error or a minor allele frequency below 5%. Association between the allele and the phenotype was assessed with Fisher’s exact test and *p*-values were corrected by Holm’s method. Population stratification was assessed by a multidimensional scaling plot of the first two principal components and a QQ plot of the uncorrected *p*-values.

## 3. Results

From 2002 to 2021, 757 kittens were born in the cattery from 43 dams and three sires. A total of 61 of these cats (8.1%), 35 males and 26 females (57% and 43% of the affected cats, respectively), were known to have developed a corneal sequestrum. The oldest of the affected cats was born in 2006, the youngest in 2019.

### 3.1. Effect of the Parents

Twenty-one dams and all three sires had at least one affected animal among their offspring. Two dams had an odds ratio significantly higher than 1 when their offspring was compared to that of all other dams: 13.0 (*p* < 0.001) for dam 1 and 2.94 (*p* = 0.011) for dam 2. Dams 2 and 3 were known to be affected; all other dams were (as far as known) healthy. The data of the 21 dams with affected offspring are shown in Table 1.

All three sires had affected offspring. Sire 1 had a significantly increased odds ratio of 2.07 (*p* = 0.014), sire 2 had an odds ratio significantly lower than 1 (0.45, *p* = 0.032) and sire 3 did not significantly deviate from the general population, as shown in Table 2.

There is a risk of confounding as dams were usually mated to only one specific sire. For example, dams 1 and 2 were always mated to sire 1 and their high odds may partially explain the increased odds for sire 1. To model the dams and sires together, the dam variable was transformed into a binary variable with dams 1 and 2 as “high risk” and all other dams as “low risk”. A logistic regression model with corneal sequestrum state in function of sire and this binary dam variable was fit. The year of birth was not included, as this was not statistically significant (*p* = 0.32), while the binary dam variable was retained because of its strong significance (*p* = 1.1 × 10^−12^). Compared to sire 3, there was no significantly higher risk for cats sired by sire 1 (*p* = 0.50) or by sire 2 (*p* = 0.20).

### 3.2. Heritability

Every prior distribution resulted in a Gelman—Rubin statistic of 1 for all parameters in the model, indicating a good convergence of the four chains. The mean posterior heritability estimates were very similar for the different priors, ranging from 0.96 to 0.98 (Table 3). The Cauchy (0, 2.5) had the widest 95% posterior interval for heritability, ranging from 0.76 to 1. The parameter coefficients for year of birth and sex were negative for all prior distributions, suggesting a higher risk for older cats and male cats. However, the 95% posterior interval for the sex parameter always firmly included zero, indicating much uncertainty around this estimate.

### 3.3. Genome-Wide Association Study

A GWAS was performed with fourteen cases and ten controls. Among the cases were dams 2 and 3 and among the controls sire 1; all other samples were from cats born from the cattery. The cases had a median age of 6 years at the time of sampling (range: 0 to 11 years) and the controls had a median age of 6 years (range: 2 to 10 years). All animals had call rates over 99%.

After filtering out variants with a low call rate (<0.95), more than one Mendelian error or a minor allele frequency <0.05, 37,595 variants remained. The lowest obtained raw *p*-value was 1.8 × 10^−4^ (Figure 1) and after correction for multiple testing, all *p*-values were adjusted to 1. Excluding the four youngest control cats (2 to 4 years at the time of sampling) decreased the corrected *p*-value of the top variant (ChrA1.146883627) to 0.24, but did not change other corrected *p*-values.

### 3.4. Breeding Decisions: Parental and Litter Phenotypes

A summary of the prevalence of corneal sequestra is provided in Table 4. As all sires were healthy, there were no litters where both parents were affected. The prevalence among progeny when one of the parents was affected (dams 2 and 3) was 22%. When the parents themselves were unaffected, the prevalence was lower, i.e., 6.9%. Exclusion of affected parents from breeding would have decreased the number of potential sires and dams to 44, i.e., 96% of the total of 46 breeding animals.

## 4. Discussion

Two out of 43 dams (4.7%) had given birth to 26 of a total of 61 affected cats (40%) and the odds of having a corneal sequestrum were significantly higher for the offspring of these two dams. The importance of the dam implies that genetic factors have a substantial influence on the development of the disease. There was no sex predisposition and the proportion of affected animals in most litters was lower than is expected for an autosomal recessive (or dominant) pattern of inheritance, unless the penetrance is very low. An autosomal dominant pattern of inheritance furthermore seems unlikely as only two out of 20 dams with affected offspring and none of the sires were known to be affected. Based on the distribution of affected cats, a complex pattern of inheritance is more likely.

The high heritability estimates further suggest that the development of corneal sequestra in cats from this cattery is mostly influenced by genetic factors and only very little by environmental factors. In theory, heritability estimates are population- and environment-specific, but in practice, heritability estimates for the same trait in different populations are often similar [23]. It is therefore likely that corneal sequestra also have a high heritability in other populations of Persian cats that suffer from this condition. A heritability close to 1 is compatible with the notion of an autosomal recessive pattern of inheritance as suggested by Vawer [16], but also with other Mendelian patterns of inheritance or a polygenic segregation [23].

A complication for the interpretation of heritability estimates is that the estimate can be inflated by a shared environment of closely related animals. In this study all animals were raised in the same cattery followed by scattering of the cats over a large number of owners at the age of 13 weeks. This makes a systematic environmental effect between litters unlikely, except for the first six weeks after birth, when the kittens stay with their mother. There is also no indication of a change in the environment over time as year of birth was not significant. If the environment in which the cats were raised underwent important changes over time, the incorporation of the year of birth into the model would have provided a correction for this.

A GWAS was conducted to search for loci that are associated with the development of corneal sequestra. No statistically significant association was found, and no single locus stood out from the others. This finding is also compatible with a complex pattern of inheritance involving multiple genetic loci that are difficult to detect in this limited sample. On the other hand, it cannot be ruled out that a major gene that influences the disease was missed because of the small sample size for the GWAS.

Segregation analysis could be used to determine which model of inheritance can best explain the observed distribution of the disease. However, software programs that can perform a segregation analysis, such as the SEGREG program in the S.A.G.E. package, cannot analyze pedigrees that contain loops. Due to multiple instances of inbreeding among the ancestors and matings of related dams with the same sire, pedigree loops were highly prevalent in this dataset. These loops could only be broken by drastically altering and reducing the pedigree. As the value and reliability of analyzing such a highly altered pedigree is questionable, no segregation analysis was performed [24].

The high heritability of corneal sequestra and the disproportional number of affected kittens from some of the parent animals imply that selection of breeding animals can have a marked effect on the prevalence of the disease in future generations. As no disease-causing variant was identified, breeders without a statistical background or access to someone who calculates estimated breeding values, will often base themselves on the phenotype of the parents. The prevalence of corneal sequestra among the offspring of unaffected dams was 6.9%, somewhat lower than the prevalence of 8.1% in all offspring, but far lower than the prevalence of 22% when one of the parents itself was affected (Table 4). Therefore, a first reduction seems to be possible by excluding affected parents. The exclusion of breeding animals with only one affected animal among many descendants or with affected distant relatives is more controversial, as excessive exclusion of cats may reduce genetic variation of the breed. For example, implementing this strategy on the pedigree at hand would result in the exclusion of all sires and 21 dams used for breeding. The late onset of the disease can be an obstacle to the successful implementation of a breeding program, as high-risk parents may only be identified after breeding. Regular ophthalmologic examination of breeding animals and breeding at a later age may somewhat mitigate this problem.

This study was limited by the reliance on reported data and lack of clinical information on some of the cats despite the fact that the breeder has contact on a regular basis with the owners of cats born in the cattery. Some owners may never report the presence of a corneal sequestrum. In the case of a discrete corneal sequestrum, few to no clinical signs can be present and may thus be unnoticed or deemed clinically unimportant by the owner. However, it seems reasonable to assume that such underreporting is independent of parentage of the cat and therefore does not substantially bias the heritability estimate or odds ratio calculations. The absence of corneal sequestra in the control cats of the GWAS was confirmed on ophthalmological examination by a board-certified veterinary ophthalmologist, with a certificate in veterinary ophthalmology, or a veterinarian with experienced ophthalmology knowledge. This makes false negative phenotypes unlikely, although incomplete and age-related penetrance of disease-causing allele(s) cannot be ruled out. To control for age-related penetrance, we repeated the GWAS with the youngest control cats excluded. This altered the results somewhat, but did not lead to statistical significance for the top variant and did not have a notable effect on neighboring variants.

On the other hand, false positive cases due to phenocopies caused by infection (FHV-1) or chronic irritation, for instance due to entropion, were also possible. Once again, it seems reasonable to assume that secondary environmental causes are independent of parentage and therefore do not substantially bias the heritability estimate or odds ratio calculations. Cases suspected to be caused by FHV-1 infection were excluded from the GWAS on the basis of their clinical history. PCR tests for herpesvirus DNA were not performed, as these are likely to be false negative for sequestrum-causing infections that have been resolved at the time of testing or may be false positive for infections that have developed or reactivated after the formation of a sequestrum. Cats with nasal entropion were not excluded as cases, as entropion is commonly seen in Persian cats. Lateral entropion of the lower eyelid was not identified in any case.

Corneal sequestration was treated as a binary variable in this study, as animals were classified as either affected or healthy. Using an ordinal classification with multiple levels of severity may allow a more nuanced investigation of the disease, but to the authors’ knowledge, an objective multigrade classification system for corneal sequestra has not yet been developed. For the purposes of genetic research, variables such as whether the cat was bilaterally affected, the age of onset or the extensiveness of the lesion (if recorded) may be used to divide the affected cats into groups of different severity.

## 5. Conclusions

Data from the cattery studied here showed that parent animals influence the odds of being affected by a corneal sequestrum in their offspring and that this disease has a high heritability. No association between the disease and a genomic locus could be established, suggesting a complex mode of inheritance.

Future studies may investigate the genetic aspects of corneal sequestra in broader and more diverse populations to see if the findings of this study can be confirmed. A GWAS with a larger sample may identify one or more loci that are implicated in the development of corneal sequestra. Using old control cats might further increase the power of a GWAS by overcoming age-related penetrance. Further phenotypical studies might help to determine the optimal age for screening and breeding. However, based on the data available, breeding advice can be provided, helping to reduce the prevalence of corneal sequestra.

## Figures and Tables

**Figure 1 animals-12-02008-f001:**
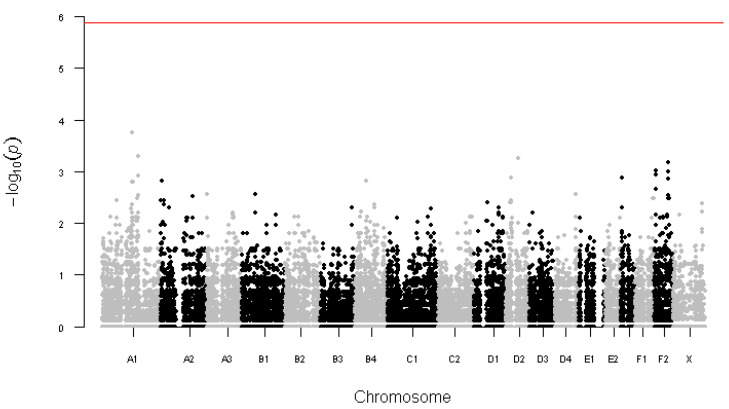
Manhattan plot of the *p*-values obtained with Fisher’s exact tests. The genome-wide significance (red line) is set at 1.3 × 10^−6^ (=0.05/37,595 tests).

**Table 1 animals-12-02008-t001:** Offspring born after 2019 of dams that had at least one affected animal among their offspring.

Dam	Number of Cases	Number of Controls	Odds Ratio	95% CI	*p*-Value
1	17	17	13.0	5.8 to 29.2	<0.001
2	9	33	2.94	1.17 to 6.7	0.011
3	4	12	3.39	0.77 to 11.7	0.053
4	1	34	0.275	0.007 to 1.70	0.24
5	1	2	4.9	0.082 to 96	0.26
6	2	11	1.80	0.189 to 8.5	0.35
7	2	11	1.80	0.189 to 8.5	0.35
8	1	4	2.45	0.049 to 25.3	0.39
9	4	27	1.47	0.362 to 4.4	0.52
10	4	28	1.41	0.349 to 4.3	0.53
11	2	37	0.51	0.058 to 2.07	0.57
12	2	15	1.31	0.142 to 5.8	0.67
13	2	17	1.15	0.126 to 5.0	0.69
14	1	17	0.57	0.013 to 3.74	1
15	1	14	0.69	0.016 to 4.7	1
16	1	18	0.53	0.013 to 3.50	1
17	1	15	0.64	0.015 to 4.3	1
18	1	13	0.75	0.017 to 5.1	1
19	1	12	0.81	0.019 to 5.6	1
20	3	36	0.80	0.153 to 2.66	1
21	1	9	1.08	0.024 to 8.0	1

CI = confidence interval. Odds ratios rounded according to Cole [22].

**Table 2 animals-12-02008-t002:** Offspring of the three sires born after 2019.

Sire	Number of Cases	Number of Controls	Odds Ratio	95% CI	*p*-Value
1	45	342	2.07	1.12 to 4.0	0.014
2	9	166	0.45	0.189 to 0.94	0.032
3	7	86	0.77	0.285 to 1.77	0.7

CI = confidence interval. Odds ratios rounded according to Cole [22].

**Table 3 animals-12-02008-t003:** Heritability estimates using four different prior distributions.

Prior Distribution	Heritability Estimate	95% Posterior Interval	Gelman–Rubin Statistic
Cauchy (0, 2.5)	0.96	0.76 to 1	1
Cauchy (0, 10)	0.98	0.88 to 1	1
Normal (0, 10)	0.98	0.86 to 1	1
Uniform (0, 10)	0.98	0.89 to 1	1

**Table 4 animals-12-02008-t004:** Prevalence of corneal sequestra in function of the parental phenotypes.

Parental Phenotype	2 Parents Affected	1 Parent Affected	2 Parents Healthy
*n* affected/total progeny (%)		13/58 (22%)	48/699 (6.9%)

## Data Availability

The pedigree and phenotype data of the cattery investigated in this study are available in Appendix A.

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
