# Peer review of "Genetic Aspects of Corneal Sequestra in a Population of Persian, Himalayan and Exotic Cats"

_animals, 2022, doi:10.3390/ani12152008_

Round 1
Reviewer 1 Report
Thank you for sending me your paper.
The aim of this study was to identify the genetic factors of corneal sequestra in Persian cats. Analysis of a relatively large sample of pedigree data in the cattery showed a high heritability, but specific genes could not be identified. However, I think it is very significant that they have put a spotlight on one disease and its genetics that has rarely been investigated.
・It is very interesting to note the high frequency of this disease in Persians and their relatives. As Persians have a very distinctive facial morphology, is there a relationship to this? Alternatively, is it possible that only this breed may have factors that make this disease severe?
・Is there any data on the general incidence rate of this disease?
・As to why they did not find any genes, the authors stated "many different genes". However, I thought that the final number of variants was low and the detection power was simply low. Or could the controls include individuals who will develop the disease in the future? In other words, what happens if younger individuals are excluded?
Furthermore, the possibility of under-reporting was mentioned. In this study, the explained variable was treated as binary (affected/not affected), but do you think it is worthwhile to include the severity of the disease as a variable in future studies? If so, it would be possible to describe it in the discussion.
Reviewer 2 Report
Dear authors,
The study investigates the genetic aspects of corneal sequestra in population of cats by GWAS analysis. The manuscript is well written and structured, the introduction provides sufficient background, the cited references are relevant to the research, the research design is appropriate, the methods are adequately described, the results are clearly presented, and the conclusions are supported by the results. Only minor format change is necessary. In tables 1 and 2, the authors should unify the number of decimals for each column.
